# Mycroft: Towards Effective and Efficient External Data Augmentation

## Abstract

Machine learning (ML) models often require large amounts of data to perform well. When the data available to the model trainer is insufficient to obtain good performance on their desired task, they may need to acquire more data from external sources. Often, useful data is held by private entities who are unwilling to share their data due to monetary and privacy concerns. This makes it challenging and expensive for model trainers to acquire the data they need to improve their model's performance. To tackle this problem, we propose `Mycroft`, a data-efficient method that enables model trainers to evaluate the relative utility of different data sources while working with a constrained data-sharing budget. Leveraging both functional and feature similarity, `Mycroft` identifies small but informative data subsets from each data owner. This allows model trainers to identify useful data owners and improve model performance with minimal data exposure. Experiments across multiple tasks in two domains show that `Mycroft` converges rapidly to the performance of the `full-information` baseline, where all data is shared. Moreover, `Mycroft` is robust to label and data noise, and can effectively recover a utility-based ranking of data owners. We believe `Mycroft` paves the way for democratized training of high performance ML models.

## 1 Introduction

Machine learning models' performance relies heavily on their training datasets, but the data available for training is often insufficient, outdated, or unrepresentative of the task (Stacke et al., 2021; Elsahar & Gallé, 2019; Liu et al., 2023). As models expand into new domains and existing ones exhaust public data, data scarcity becomes a key challenge. According to economic and manufacturing experts, one of the primary reasons that AI technology is not being widely adopted in manufacturing is the lack of relevant public data available for production tasks (Alam et al., 2024). While large corporations and governments can afford large-scale data collection or use methods like crowdsourcing (Sigurdsson et al., 2016) and federated learning (Kairouz et al., 2021) to increase their access to diverse datasets, entities without such resources, such as individuals and small businesses, find it extremely challenging to collect their own data.

Such entities may need to acquire data from private entities (referred to as *data owners*), who are often unwilling to share it publicly, especially when it involves proprietary or sensitive data, such as educational or health records (Spector-Bagdady et al., 2019). Acquiring data from these private data owners can involve high overhead, such as the creation of complex data sharing agreements, monetary compensation, and compliance with regulations such as (GDPR, 2021). Therefore, data sharing protocols which help model trainers assess which data owners are most likely to provide useful data before acquiring data at scale are needed.

**Challenges of external data augmentation:** The model trainer needs to efficiently select one or more data owners whose datasets match their requirements. In the case with a profusion of data owners, while a large number can be filtered out just on the basis of metadata (*i.e.* domain, collection methodology, licensing requirements etc.), it is unclear which of the remaining data owners the model trainer should acquire data from. Even in the case of a single data owner with a large pool of data, both the model trainer and data owners will benefit from a method to assess the dataset's potential effectiveness. In summary, the key question we aim to answer in this paper is:

*How can a model trainer assess the usefulness of data owners without acquiring their entire dataset?*

**Mycroft for external data augmentation:** We present Mycroft, a framework that helps the model trainer and data owners identify relevant data (from the data owners) for improving the model trainer's performance. The framework follows three main steps: (1) the model trainer sends information about the task on which their model is under-performing, typically through data samples, to data owners; (2) the data owners use an efficient algorithm to identify a small, relevant subset of their data to demonstrate their dataset's usefulness and send it to the model trainer; and (3) the model trainer evaluates this subset to decide whether to acquire additional data from the data owners.

The *key technical challenge* we solve in this paper lies in Step (2), where each data owner must find the most "relevant" training data with respect to the model trainer's transmitted data samples, which may not follow the same distribution as their training data. Our proposed algorithms select data based on either functional similarity, which compares sample-wise gradients from task-specific models, or feature similarity, which measures the distance between samples in a relevant feature space, or a combination of both. In summary, our contributions are as follows:

**1. Modelling the problem of external data augmentation (§2:)** We formally model the practical problem of acquiring data from private data owners by stating key assumptions and adding constraints inspired by the likely real-world operating conditions for any external data augmentation algorithm.

**2. Mycroft: a protocol for identifying and sharing useful data (§3:)** We outline a data sharing protocol, which crucially depends on methods for the data owner to identify relevant data. We propose two data selection approaches, based on functional (loss gradient) similarity and feature similarity, and highlight their applicability. We adapt existing tools to make them suitable for our goal and in cases where these techniques were ineffective (such as for tabular data), we develop a new metric for data selection. Finally, we unify both methods via a joint optimization objective to leverage the notions of similarity arising from each approach.

**3. Extensive methodological assessment (§4:)** We compare Mycroft against two baselines: (i) full-information, where each data owner shares all their data with the model trainer, serving as the optimal baseline, and (ii) random uniform sampling. Across multiple classification tasks from 2 domains (computer vision and tabular data) and a range of data budgets, our experiments show that Mycroft significantly outperforms random sampling and quickly approaches the performance of full-information under a much smaller budget. In the vision domain, Mycroft outperforms random-sampling by an average of 21% for five data budgets on four datasets. In the tabular domain, Mycroft with just 5 samples outperforms random-sampling with 100 samples and matches full-information performance for 65% of sharing scenarios.

**4. Case studies of using Mycroft in practical settings (§5):** We conduct four additional case studies to evaluate Mycroft's effectiveness across different scenarios that may be encountered in practice. We show that Mycroft is robust to instance and label noise in the data provided by the data owners. When this data lacks labels entirely, feature similarity is still effective at finding relevant data samples. When multiple data providers are involved, Mycroft successfully reconstructs a utility-based preference order that closely aligns with the full-information setting. These findings confirm that Mycroft offers a highly data-efficient solution for data owners to reliably demonstrate the value of their data to a model trainer, even in noisy or complex environments.

We hope Mycroft and its associated open-source code [1] paves the way for more efficient and private data-sharing frameworks, in turn helping democratize the training of performant ML models.

## 2 PROBLEM SETUP AND FORMULATION

In this section, we first present the challenges associated with external data augmentation in practical settings and the desired properties for any proposed method. We then formally define the specific problem we solve and present the required notation.

### 2.1 CHALLENGES OF EXTERNAL DATA AUGMENTATION

The most direct approach to external data augmentation is to acquire all data from all available data providers. However, this approach presents the following challenges:

---

[1]Anonymize code is available at : https://anonymous.4open.science/r/Mycroft-73FE/

1. **Data owners are unwilling to share all their data:** Due to privacy and proprietary concerns, data owners are often unwilling to share all their data. They also typically expect compensation which necessitates the use of data budgets to establish the utility of the dataset before sharing.

2. **Sharing poorly curated data can negatively impact performance:** A large portion of the data owned by data owners may be poorly curated or irrelevant for model training. Including such data in the training process can degrade model performance, so it is crucial for model trainers to select only high-quality and relevant subsets from the data owners' contributions.

3. **Data budgets can ease computational concerns:** Sharing large volumes of data from multiple owners makes it difficult and costly for model trainers to update their models frequently.

## 2.2 Problem Formulation and Notation

We consider a setting where there is a model trainer (MT) and $m$ data owners (DOs). MT has trained a model $M_{\text{MT}}$ on its own dataset $D^{\text{MT}}$ and is aiming to improve their performance (or lower their loss) on test data $D^{\text{test}}$ via external data augmentation. The performance of MT's model on $D^{\text{test}}$ is measured with respect to some task. In this paper, we assume that the task is supervised learning, so the overall performance is measured as $L(D^{\text{test}}, M) = \sum_{z_i \in D^{\text{test}}} \ell(y_i, M(x_i))$, where $z_i = (x_i, y_i)$ is a labeled sample and $\ell(\cdot, \cdot)$ is some appropriate loss function such as cross-entropy loss.

We then posit that there exists a subset $D^{\text{hard}}$ of $D^{\text{test}}$ on which MT is aiming to improve their performance, leading them to use external data augmentation. We explain how $D^{\text{hard}}$ is constructed in specific settings in 4.1. Each DO has a dataset $D_i$ which could aid MT in improving their performance on $D^{\text{hard}}$. However, due to the challenges highlighted in §2.1, the DOs do not share all of their data with MT. Rather, they share a small subset $D^{\text{useful}}$ of up to size $k$, which we call the *budget*. If $D^{\text{useful}}$ is able to improve the performance of MT's model, then MT and DO could potentially enter into a data-sharing agreement for additional data acquisition. This paper focuses on how each DO can identify $D^{\text{useful}}$ and subsequently, how MT can utilize this data to improve performance and if $m > 1$, rank the DOs. The task for each DO is then:

**Definition 2.1** (Task for each DO). Find $D_i^{\text{useful}} \subseteq D_i$ such that $|D_i^{\text{useful}}| \leq k$ and $L(D^{\text{hard}}, M'_{\text{MT}}) \leq L(D^{\text{hard}}, M_{\text{MT}})$, where $M_{\text{MT}} = \mathcal{D}(D^{\text{MT}})$ and $M'_{\text{MT}} = \mathcal{D}(D^{\text{MT}} \cup D_i^{\text{useful}})$.

We summarize our key assumptions below about the data held by various entities:

**A.1** There exists a subset $D^{\text{hard}} \subseteq D^{\text{test}}$ with accurate ground-truth labels on which $M_{\text{MT}}$ performs poorly, which MT shares with the DOs.

**A.2** Each participating DO has samples from at least one of the classes contained within $D^{\text{hard}}$.

We assume **A.1** because if MT does not share any knowledge of the difficult subset, DOs cannot share meaningful data. Further, if $D^{\text{hard}}$ is incorrectly labeled, basic challenges regarding performance evaluation arise. **A.2** just rules out DOs with no relevant data to the task under consideration.

Additionally, our algorithm accounts for the following constraints likely to be encountered in practice:

**C.1** MT does not share $M_{\text{MT}}$ with the DOs.

**C.2** No DO shares their full training data with MT, *i.e.* $\forall i, k < |D_i|$.

**C.1** addresses the reality that the model trainer is unlikely to share their local model for intellectual property and privacy reasons. In spite of the added challenge, we show in §4 that the DOs can effectively share useful data. Additionally, **C.2** arises from the fact that DO will provide just enough data to convince MT of their data utility due to privacy and economic concerns. After utility is established, MT may enter into an agreement with the best DO (s).

## 3 Mycroft: Identifying and sharing useful data

In this section, we present Mycroft, our data sharing protocol between the MT and DOs.

**Overview of approach:** The overall approach is outlined in Algorithm 1. In brief, after each DO receives the dataset $D^{\text{hard}}$ from MT, they will use it to identify a subset $D_i^{\text{useful}}$ of size $k$ from their local training data $D_i$ that is relevant for MT to predict $D^{\text{hard}}$ correctly. To do so, DO can use

---

**Algorithm 1** Mycroft

---

**Require:** $M_{\text{MT}}$, $D^{\text{test}}$, DO's loss function $L_{\text{DO}}$, $\text{DO}_i$'s dataset $D_i$, Budget $k$,
1: $D^{\text{hard}} \leftarrow \mathcal{D}_{\text{test}}(D^{\text{test}}, M_{\text{MT}})$
2: Send $D^{\text{hard}}$ to the DO$s$
3: $D_i^{\text{useful}} \leftarrow \text{DO}_i$ runs DataSelect     ▷ DataSelect calls **OMP** 2 or **FeatureSimilarity** 3
4: $M'_{\text{MT}} = \mathcal{D}(D^{\text{MT}} \cup D_i^{\text{useful}})$.           depending on whether $L_{\text{DO}}$ is differentiable

---

either functional similarity (loss gradient similarity) or feature similarity, or a combination of both to identify $D_i^{\text{useful}}$, depending on whether the loss function $L_{\text{DO}}$ is differentiable. The DO then sends $D_i^{\text{useful}}$ to MT, who will use this data to update their model $M_{\text{MT}}$. The key technical challenge we address in this section is that of designing the subroutine DataSelect.

### 3.1 FUNCTIONAL SIMILARITY VIA LOSS GRADIENT MATCHING

For models trained using a differentiable loss function, the gradient of the loss with respect to each model parameter shows how each sample affected the model during training. Samples that produce similar gradients are considered functionally similar. This concept has been used in several studies to sub-select training data to improve the efficiency of model training (Mirzasoleiman et al., 2020; Killamsetty et al., 2021a). Ideally, to find the most relevant samples to $D^{\text{hard}}$, the model used would be $M_{\text{MT}}$. However, due to constraint **C.1** that no DO has access to MT's model, we assume that each DO has a model $M_i$ with parameters $\theta_i$ trained using a differentiable loss function $L$ that can function as a reasonable proxy (our empirical findings show that this method works well in practice).

We formulate the problem of finding $D_i^{\text{useful}}$ as that of obtaining a $k$-sparse weight vector $\mathbf{w}$ over $D_i$, with the weight assigned to each sample corresponding to its utility. To find this $k$-sparse $\mathbf{w}$, we: (1) find the averaged gradient of the loss $L$ computed on $D^{\text{hard}}$ with respect to the parameters $\theta_i$ of $M_i$ (denoted $\nabla_{\theta_i} L(D^{\text{hard}})$); (2) compute the gradient of the loss $L$ computed on each sample $z_j \in D_i$ with respect to the parameters $\theta_i$ of $M_i$; (3) solve the following regularized optimization problem:

$$\min_{\|\mathbf{w}\|_0 \leq k} e_\lambda(\mathbf{w}) = \min_{\|\mathbf{w}\|_0 \leq k} \left\| \sum_{z_j \in D_i} \mathbf{w}_j \nabla_{\theta_i} L(z_j) - \nabla_{\theta_i} L(D^{\text{hard}}) \right\| + \lambda \|\mathbf{w}\|_2^2. \tag{1}$$

The first term in Eq. 1 ensures that a weighted sum of the selected samples is close to the gradient of the loss on $D^{\text{hard}}$, while the regularization term prevents the assignment of very large weights to a single instance. The $\ell_0$-"norm" constraint on $\mathbf{w}$ enforces sparsity but leads to an NP-hard problem. To tackle this issue, we can use a greedy algorithm, Orthogonal Matching Pursuit (OMP) (Pati et al., 1993), to find a close approximation due to the sub-modularity of $e_\lambda(\mathbf{w})$ (Elenberg et al., 2016) [2]. We detail OMP in Algorithm 2. The choice of which state of the model to use (typically stored as checkpoints) is explored in §4.3.

**Other potential gradient based techniques:** Before we settled on utilizing OMP on the model gradients for the DataSelect process, we also explored various other coreset techniques (Zhou et al., 2022; Coleman et al., 2019; Toneva et al., 2018; Ducoffe & Precioso, 2018; Ren et al., 2018). Since these techniques are designed to create subsets of data that can approximate the loss of an entire dataset, most of them are not directly applicable to our setting which requires finding a small subset which will be useful for a specific task to the MT. Additionally, these techniques are designed to be used jointly when models are trained from scratch whereas we are only interested in efficient solutions that can find useful samples by leveraging an already trained model. Among coreset techniques that can be adapted to our problem setting, we decide to use OMP because other techniques either underperform our chosen approach (Mirzasoleiman et al., 2020) or are too computationally intensive to be feasible (Ren et al., 2018).

---

[2]This use of OMP is inspired by (Killamsetty et al., 2021a), who use it for dataset compression during training.

## 3.2 Feature Similarity

We also find that data samples which are similar to samples from $D^{\text{hard}}$ in an appropriate feature space are useful for MT to augment their training set. This is especially useful for models which have been trained *without differentiable loss functions*. Given any good feature extractor $\phi(\cdot)$ that maps a sample $x_j$ to its feature representation $\phi(x_j)$, we compute the distances of each sample in $D^{\text{hard}}$ to each sample in $D_i$ and store them in a matrix $\Psi \in \mathcal{R}^{|D^{\text{hard}}| \times |D_i|}$. We construct $D_i^{\text{useful}}$ by using a greedy heuristic which first sorts this matrix in the row dimension followed by the column dimension. We select the top-$k$ samples by iterating over the columns which selects $|D_i^{\text{useful}}|$ samples with the minimum distance to $D^{\text{hard}}$ samples while ensuring coverage of $D^{\text{hard}}$ (see Algorithm 3 for more details). The choice of feature representations $\phi(\cdot)$ and distance function $d(\cdot, \cdot)$ is contingent on the domain of the data and the classification task. For the image datasets, we use the feature space of a image retrieval model called Unicom (An et al., 2023) and $L_2$ distance as our distance function. For the tabular dataset where existing techniques were ineffective, we propose a `ExtractBinningFeatures` algorithm using Hamming distances over adaptive grids (Appendix 4).

## 3.3 FuncFeat : Combining Gradient and Feature Similarity

Whenever the DO has access to a model trained on $D_i$ as well as a good feature extractor, they can combine both notions of similarity to find useful samples. Combining both notions of similarity may improve the quality of $D_i^{\text{useful}}$ as samples which can align both in the feature and gradient space are more likely to be relevant to $D^{\text{hard}}$. Our technique (`FuncFeat`) introduces a regularization term in terms of a composite norm that incorporates the feature similarity between samples from $D^{\text{hard}}$ and $D_i$ to the approximation error from Eq. 1:

$$e'_{\lambda_1, \lambda_2}(\mathbf{w}) = e_{\lambda_1}(\mathbf{w}) + \lambda_2 \|\Psi \mathbf{w}\|_2^2, \tag{2}$$

where $\Psi$ is a matrix of distances. The second term functions as a regularizer that penalizes samples that are far from feature representations of $D^{\text{hard}}$. In the following theorem, we show sub-modularity:

**Theorem 3.1.** *If the loss function $L(\cdot)$ is bounded above by $L_{max}$ and $\forall j, \|\nabla_{\theta_i} L(z_j)\| \leq \nabla_{max}$, then $f_{\lambda_1, \lambda_2}(\mathbf{w}) = L_{max} - e'_\lambda(\mathbf{w})$ is weakly submodular with parameter $\gamma' \geq \frac{\lambda_1 + \lambda_2 \|\Psi\|_2^2}{\lambda_1 + \lambda_2 \|\Psi\|_2^2 + k \nabla_{\max}^2}$,*

where $\|\cdot\|_2$ is the spectral norm for matrices. From Elenberg et al. (2016), we get that OMP returns a $1 - e^{\gamma'}$-close approximation of the maximum value of $f_{\lambda_1, \lambda_2}(\mathbf{w})$.

## 4 Experiments

In this section, we empirically demonstrate that `Mycroft` outperforms `random-sampling` and rapidly approaches `full-information` while operating within the communication constraints laid out in §2. We also present several ablation studies to investigate key design choices for `Mycroft`.

### 4.1 Experimental Setup

In this section, we describe details about the datasets, models, and metrics we use to evaluate `Mycroft`, followed by how we construct $D^{\text{hard}}$ and evaluate `Mycroft`.

**1. Datasets:** We evaluate `Mycroft` on classification tasks over two domains: computer vision and network traffic classification. For the computer vision tasks, we use six different datasets, while for the network traffic classification, we use a tabular dataset that represents flow features of network traffic. Further details are in Appendix D.

**Image datasets:**

- **Food datasets:** We use three datasets from the food computing domain: Food-101 (Bossard et al., 2014), UPMC Food-101 (Wang et al., 2015) and ISIA Food-500 (Wang et al., 2015). We assign Food-101 to be the MT's dataset and UPMC Food-101 and ISIA Food-500 as datasets of two DOs.
- **Dog datasets:** We use two datasets for dog breed classification: Imagenet-Dogs (Deng et al., 2009) and Tsinghua-Dogs (Zou et al., 2020). Imagenet-Dogs contains 120 dog classes from Imagenet

| Budget $k$ | Food-101 - UPMC | | Food-101 - ISIA500 | | Imagenet - Tsinghua | | Dogs & Wolves - Spurious - Dogs & Wolves - Natural | |
|---|---|---|---|---|---|---|---|---|
| | Mycroft | random-sampling | Mycroft | random-sampling | Mycroft | random-sampling | Mycroft | random-sampling |
| 8 | 0.42 | 0.36 | 0.33 | 0.18 | 0.43 | 0.31 | 0.44 | 0.13 |
| 16 | 0.50 | 0.38 | 0.48 | 0.33 | 0.62 | 0.48 | 0.50 | 0.13 |
| 32 | 0.61 | 0.47 | 0.54 | 0.40 | 0.70 | 0.56 | 0.75 | 0.25 |
| 64 | 0.75 | 0.64 | 0.72 | 0.61 | NA | NA | 0.75 | 0.25 |
| 128 | 0.86 | 0.72 | 0.83 | 0.70 | NA | NA | 0.88 | 0.38 |

Table 1: Accuracy of $M'_{\mathrm{MT}}$ with varying budgets of $D_i^{\mathrm{useful}}$. `full-information` results in 100% accuracy on $D^{\mathrm{hard}}$. Column headers indicate the `MT` and `DO` datasets separated by hyphens.

whereas Tsinghua-Dogs contains 130 dog classes which include all the classes in Imagenet-Dogs. We consider Imagenet-Dogs and Tsinghua-Dogs to be the `MT`'s and `DO`'s dataset respectively.

- **Dogs & Wolves:** We curate a dataset of Dogs and Wolves with spurious correlations to simulate a controlled `MT-DO` interaction, which helps illustrate effectiveness of `Mycroft`. The `MT` model is trained on a dataset containing these spurious correlations, referred to as Dogs & Wolves - Spurious, therefore, the model performs poorly on a dataset without these correlations. We label such a dataset as Dogs & Wolves - Natural and simulate a `DO` which has this dataset. An illustration of this dataset is provided in Figure 5 in the Appendix.

**Tabular dataset:** We use the IoT-23 dataset (Garcia et al., 2020) which contains tabular features derived from the network traffic flows. The `MT`'s task is to select `DO`s that would improve its model's ability to detect malicious traffic. In order to make the evaluation comprehensive, we experiment with different combinations of `MT`s, `DO`s, and types of malicious attacks totalling 665 combinations.

**2. Models:** For the image datasets, all our experiments use ResNet50 (He et al., 2016) models pre-trained on Imagenet since after evaluating the vision datasets on several CNN architectures, including MobileNets, EfficientNets, and ResNets (18 & 50), we found that our results were consistent across these architectures. For the tabular dataset, we use Decision Trees, XGBoost and Random Forest as previous works (Grinsztajn et al., 2022; Hasan et al., 2019; Yang et al., 2022) have shown that simple models are not only computationally effective but also can outperform deep learning models for typical tabular datasets. Appendix D.2 contains details for the training procedure.

**3. Metrics & Baselines:** We define `full-information` to be the setting where `MT` uses a `DO`'s entire dataset to train their model. This represents the upper bound on the performance improvement with external data augmentation from a `DO`. We use `random-sampling` as our baseline technique for external data augmentation with knowledge of the class label from which data is to be retrieved. Several works have shown that random sampling is an effective strategy for dataset compression (Mahmud et al., 2020; Guo et al., 2022; Mirzasoleiman et al., 2020; Killamsetty et al., 2021b), which is highly relevant to our task and thus a valid baseline. For the image datasets, we report classification accuracy (normalized between 0 and 1) and F1 score for the tabular dataset (due to class imbalance) for $D^{\mathrm{hard}}$.

**4. Construction of $D^{\mathrm{hard}}$:** Our approach for creating the $D^{\mathrm{hard}}$ subset involves two steps. First, we identify the misclassified samples in the validation set, $D^{\mathrm{val}}$, forming $D^{\mathrm{hard}}$. Then, if $D^{\mathrm{hard}}$ is large enough, we randomly select a small subset to share with the `DO`s and keep the remaining for evaluation. If it is too small to meaningfully split, we share the entire $D^{\mathrm{hard}}$.

**5. Evaluation setup:** For the tabular dataset, where we have enough samples for a meaningful split, we evaluate the performance of the `MT`'s model on a separate test dataset not seen by `MT`. In vision tasks, where $D^{\mathrm{hard}}$ often lacks sufficient samples for such a split, we evaluate the model directly on the $D^{\mathrm{hard}}$ dataset. When a meaningful split is possible on $D^{\mathrm{hard}}$, we evaluate on the held-out $D^{\mathrm{hard}}$ data. For evaluation, whenever model gradients are available, we use our *FunctFeat* technique to construct $D_i^{\mathrm{useful}}$. Otherwise, we use our feature similarity methods.

## 4.2 RESULTS

Here, we present the results of evaluating `Mycroft` on different datasets and compare it with our `random-sampling` baseline. We also vary data budgets to understand how well `Mycroft` can approximate the performance of the `full-information` setting.

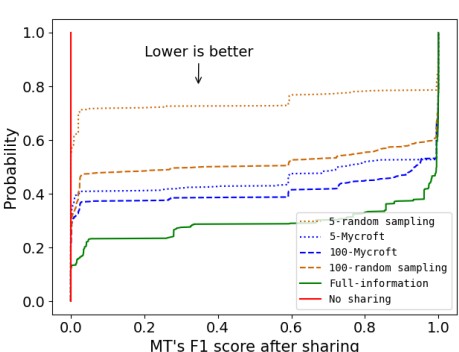 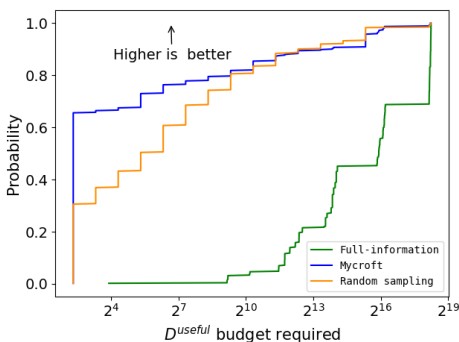

(a) CDF of `MT`'s F1 score after data sharing. As the number of samples in $D_i^{\text{useful}}$ increases, both the performance for `random-sampling` and `Mycroft` increase and move closer towards `full-information` performance. `Mycroft` outperforms `random-sampling`, as can be seen from the fact that `Mycroft` with $D_i^{\text{useful}}$ of 5 samples outperforms `random-sampling` with $D_i^{\text{useful}}$ of 100 samples.

(b) CDF of $D_i^{\text{useful}}$ budget required for `random-sampling` and `Mycroft` to match `full-information` for cases where sharing data is helpful using DecisionTree classifier. For most cases, `Mycroft` uses a much smaller $D_i^{\text{useful}}$ budget compared to `random-sampling` to match the performance of `full-information`. N = 474 cases where sharing data is helpful (F1 score for `full-information` >= 0.5).

Figure 1: Performance of `Mycroft` compared to `random-sampling` and `full-information` on the tabular dataset.

**Image Datasets:** We present results for the image datasets Table 1 and make three key observations. First, we can see that `Mycroft` outperforms `random-sampling` across all datasets and at all budgets of $D_i^{\text{useful}}$. Secondly, we note that `Mycroft` can rapidly converge to the `full-information` setting using only a fraction of the dataset. To be specific, the highest budget on $D_i^{\text{useful}}$ amounts to training on at most **32%** of the `DO`'s dataset, on average, in our experiments, yet `Mycroft` is able to reach at least **83%** of the `full-information` performance.

**Tabular Dataset:** Fig 1a displays the result for the tabular dataset. We see that $D_i^{\text{useful}}$ budget of 5 samples retrieved using `Mycroft` outperforms $D_i^{\text{useful}}$ of 100 samples retrieved using `random-sampling`. In addition, as we increase $D_i^{\text{useful}}$ budget to 100, we move closer to the `full-information` performance. To fully explore the performance of `Mycroft` on various budgets, we start with a $D_i^{\text{useful}}$ budget of 5 samples and double it until the `MT`'s performance reaches that of the `full-information` setting (Fig 1b). We observe that `Mycroft` reaches the `full-information` performance using a smaller $D_i^{\text{useful}}$ budget compared to `random-sampling`, especially on small $D_i^{\text{useful}}$ budgets. For example, with a budget of merely 5 samples, `Mycroft` can reach the same performance as `full-information` on **65%** of the combinations used, as opposed to **30%** for `random-sampling`. Thus, `Mycroft` significantly reduces the amount of data that needs to be shared to obtain the same improvement in performance as `full-information`. For this dataset, we note that `Mycroft` only uses feature similarity to retrieve useful data. Further discussion in is Appendix E.

### 4.3 ABLATION STUDIES

We conduct ablation studies on the image datasets to analyze key design choices of `Mycroft`. For tabular dataset ablations, see Appendix E.2.

**Fine-tuning vs training from scratch:** We compare two approaches for the `MT` to incorporate $D_i^{\text{useful}}$ into their model: finetuning and retraining. We find that retraining from scratch and fine-tuning both achieve similar performance but fine-tuning is much more computationally efficient. In particular, training `MT`'s models with externally augmented data from scratch takes 8 hours on average on a single Nvidia A40 GPU whereas finetuning only requires approximately 50 minutes. Hence, we choose to use finetuning for all our experiments.

**Checkpoint selection for loss gradient matching:** We find that earlier checkpoints provide more useful gradients for gradient matching (Figure 6). Please refer to Appendix E for more details.

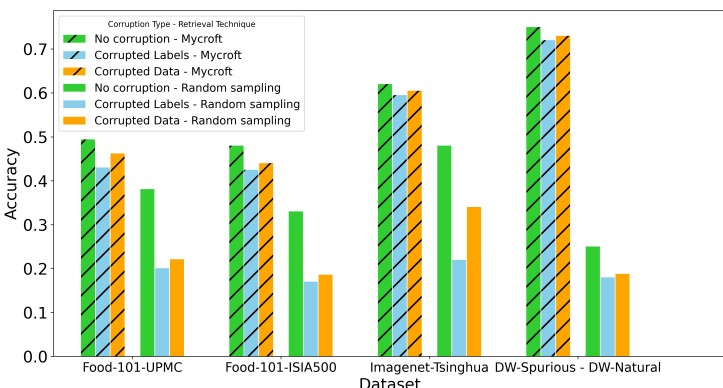

Figure 2: Accuracy of $M'_{\mathrm{MT}}$ when trained on $D_i^{\mathrm{useful}}$ retrieved using `Mycroft` and `random-sampling` under the scenario where approximately 70% of the data or labels are corrupted.

**Gradient matching with different `MT-DO` model architectures:** We also explore over the impact on the performance of gradient matching when the `DO` and `MT` have different architectures. Concretely, we keep the `MT`'s architecture as ResNet50 but change the `DO`'s architecture to EfficientNetB0 (Tan & Le, 2019). Overall, we observe small differences in the performance of the augmented model ($< 0.01$) but note that gradient matching over different architectures (Jain et al., 2024) is an avenue for future exploration.

**Other feature spaces for similarity matching:** Besides Unicom, we also explore other feature spaces, such as that of a pretrained ResNet50 . We evaluate feature spaces on the Dogs & Wolves dataset and measure how many useful samples the `DO`'s model can retrieve from the Dogs & Wolves - Natural dataset when performing similarity matching in each feature space. We observe that 76% of retrieved samples are useful when using Unicom, whereas only 6% are useful when using the ResNet50 feature space. We show a sample of the Top-k retrieved samples in Figure 4.

## 5 CASE STUDIES

In this section, we examine the applicability of `Mycroft` across different real-world data-sharing scenarios. The first three scenarios are evaluated using datasets. all our vision datasets, while the last scenario includes results for both the vision and tabular datasets.

**Scenario 1 - Corrupted data features:** In real-world scenarios, data features can be corrupted due to factors like hardware failures as well as collection and transmission errors. To assess `Mycroft`'s reliability in this context, we corrupt the `DO` dataset using random image transformations, applying random masking, color jitters, etc. The results, shown in Figure 2, indicate that `Mycroft`'s performance declines by only 2.7% on average, compared to a 13.7% drop for `random-sampling`. This demonstrates `Mycroft`'s resilience to corrupted data while still retrieving useful subsets.

**Scenario 2 - Corrupted labels:** Large-scale supervised learning datasets often contain incorrect labels. To evaluate `Mycroft`'s performance in such conditions, we randomly permuted 70% of the labels in `DO`'s dataset. The results, shown in Figure 2, demonstrate that `Mycroft` is highly robust to label corruption, with an average performance drop of only 4.4%, compared to a 16.9% decrease for `random-sampling`. This highlights `Mycroft`'s effectiveness in handling noisy labels.

**Scenario 3 - `Mycroft` in missing label settings:** In cases where the `DO` dataset lacks labels entirely, `Mycroft` can rely on feature-space distances. For vision datasets, `Mycroft` can use feature distances (Unicom) to retrieve samples and assign them pseudo-labels by matching each sample to the closest $D^{\mathrm{hard}}$ sample in the feature space. The results in Table 2 show minimal performance degradation between the scenario where labels are available (gradient similarity can be used) and not available (only feature similarity can be used).

**Scenario 4 - Preference ordering for several `DOs`:** In data markets, multiple sellers often provide datasets with varying utility. In such cases, `Mycroft` should be able to rank these datasets to support

| MT - DO | Mycroft No labels | random-sampling No labels | Mycroft With labels |
|---|---|---|---|
| Food-101 - UPMC | 0.58 | 0.19 | 0.61 |
| Food-101 - ISIA500 | 0.47 | 0.14 | 0.54 |
| Imagenet - Tsinghua | 0.44 | 0.25 | 0.7 |
| DvW - Spurious - DvW - Natural | 0.68 | 0.21 | 0.75 |

Table 2: Accuracy of `Mycroft` compared to `random-sampling` for **Scenario 3** where the `DO` has an unlabelled dataset. In this case, `Mycroft` utilizes feature similarity to construct $D_i^{\text{useful}}$. We compare the accuracy in this scenario with the performance of `Mycroft` when the `DO` has labels and utilizes gradient similarity.

| DO | Mycroft | random-sampling | full-information |
|---|---|---|---|
| DO-1 | 0.81 | 0.18 | 0.88 |
| DO-2 | 0.63 | 0.31 | 0.69 |
| DO-3 | 0.44 | 0.25 | 0.63 |
| DO-4 | 0.56 | 0.25 | 0.50 |
| DO-5 | 0.19 | 0.25 | 0.13 |

Table 3: Preference ordering (based on accuracy) generated from `Mycroft` and `random-sampling` for selecting from among several `DO` candidates with different levels of utility (where each `DO`'s number corresponds to their utillity) from **Scenario 4**. `Mycroft` is mostly able to retrieve the ground-truth preference ordering whereas `random-sampling` fails to do so.

| MT | Mycroft | random-sampling | full-information |
|---|---|---|---|
| MT-1 | 49 | 34 | 80 |
| MT-2 | 16 | 5 | 40 |
| MT-3 | 76 | 58 | 92 |
| MT-4 | 87 | 23 | 90 |
| MT-5 | 60 | 37 | 60 |
| MT-6 | 0 | 0 | 24 |
| MT-7 | 92 | 25 | 88 |

Table 4: Number of useful DOs (F1 score of $M'_{\text{MT}}>=0.5$) retrieved by `Mycroft` and `random-sampling` for budget of 5 samples and by `full-information` for different `MT`s for tabular data. Number of `DO` candidates = 95.

more informed data-sharing agreements. We tested this by constructing several `DO` datasets with different utility levels (details in Appendix F) to see if `Mycroft` could correctly rank them. The results, shown in Table 3, indicate that `Mycroft` successfully identifies the most promising datasets, while `random-sampling` struggles to do so.

For tabular data, where clear rankings are harder to establish (useful datasets often perform similarly), we measured the number of useful `DO`s (defined as those achieving an F1 score of 0.5 or higher after data sharing) retrieved by `Mycroft` and `random-sampling`. As shown in Table 4, `Mycroft` consistently retrieves more useful datasets across different `MT`s, demonstrating its ability to capture the true utility of various datasets. Notably, `Mycroft` sometimes even outperforms `full-information` in selecting useful data, as discussed in Appendix F. This highlights `Mycroft`'s suitability for ranking datasets in data-sharing scenarios.

# 6 RELATED WORK

**Data augmentation and synthetic data generation:** Numerous studies have investigated ways to enhance training data quality by leveraging existing datasets through methods such as image overlay (Inoue, 2018), random erasure (Zhong et al., 2020), and common data augmentation techniques like rotation and cropping (Inoue, 2018; Zhong et al., 2020; Chlap et al., 2021; Shorten & Khoshgoftaar, 2019; Hussain et al., 2017; Feng et al., 2021). These strategies are designed to generate new samples, reduce overfitting, and improve model generalization. However, they are less effective if the test data deviates significantly from or is underrepresented in the training set. Similarly, using generative models to create additional training samples (Tripathi et al., 2019; Such et al., 2020; Jiang et al., 2024) can enhance performance, but this approach often requires substantial data to train the generative model itself. Moreover, synthetic data may fail to capture the full complexity of real-world data distributions, limiting its impact on hard sample performance.

**Data augmentation using publicly available data:** Researchers have also explored sourcing data from publicly available data lakes (Nargesian et al., 2023; Castelo et al., 2021; Yakout et al., 2012; Esmailoghli et al., 2021; Santos et al., 2021; Zhu et al., 2016; Castro Fernandez et al., 2019; Fernandez, 2018; Galhotra et al., 2023). For example, METAM (Galhotra et al., 2023) profiles various datasets to identify and select the most relevant ones, thereby improving the quality of training data and enhancing performance on downstream tasks. Another approach, Internet Explorer (Li et al., 2023), retrieves task-specific images from the internet to support self-supervised learning. However, these solutions rely on open access to data and are ineffective in scenarios where data access is restricted. Consequently, they do not address the challenge of sourcing data from private entities.

**Selecting data without full information:** Recent work called Projektor (Kang et al., 2024) addresses the problem of selecting and weighting useful data owners without full information about the underlying data. Projektor assumes that some of the `DO`'s data is publicly available as pilot data and can predict the usefulness of the entire dataset based on this subset. This assumption may not be true in our context as the publicly available pilot data is not selected based on any knowledge about the `MT`'s task, and may be irrelevant for `MT`. Techniques like Projektor can complement `Mycroft` by finding the best combination of relevant samples from different `DO`s.

**Deriving coresets for large datasets:** There is also extensive research on identifying useful subsets of datasets, such as work on "coresets" (Killamsetty et al., 2021c; Kim & Shin, 2022; Guo et al., 2022), which focus on selecting a representative subset that approximates the cost function of the entire dataset. However, most of these methods are better suited for dataset compression rather than creating task-specific subsets. In our context, approximating the entire cost function is irrelevant to the model trainer's needs. Instead, we prioritize selecting a subset most relevant to the specific task and have adapted some of these techniques accordingly.

# 7 DISCUSSION

**Why does `Mycroft` help?** The fact that `Mycroft` outperforms `random-sampling` in almost all scenarios and quickly approaches `full-information`, even when operating under a limited data budget has scenario-dependent explanations. First, for many `DO`s, the majority of their data is irrelevant to the `MT`'s task, as such, `random-sampling` is inefficient and sharing all their data is unnecessary. Secondly, in many cases, `MT` may only need a small subset of the data to drastically improve their model's performance. For example, in the Dogs & Wolves dataset, only a small subset of the data is needed break the spurious correlations and improve generalization. Similarly, for the IoT attack data, the significant distinction between attack distributions implies a small subset is sufficient to differentiate between benign and malicious traffic, and different kinds of malicious traffic.

**Limitations and Future Work:** One limitation of our work is that it assumes that the model trainer already has access to a set $D^{\text{hard}}$ that they want to improve on. This may often be a reasonable assumption because model trainers are looking to improve on the predictions of $D^{\text{hard}}$ that they perform poorly on or have low confidence about. However, the case where $D^{\text{hard}}$ is too limited and therefore, does not give the data owners enough of a signal to determine useful data, is an important for future work. In addition, our method also assumes that the most useful data to share are likely those that are similar to $D^{\text{hard}}$. In reality, data that are not close to $D^{\text{hard}}$ might also be useful to share. In this paper, we do leverage functional similarity via gradient matching to include some diverse data points. However, explicitly promoting diverse data selection is a key direction for future work.

Finally, there remains the issue that model trainers may not be willing to share even small subsets of their data with the data owners, in which case more privacy-preserving methods can be explored on top of `Mycroft`. Future directions to address the privacy limitations include developing a version that avoids directly sharing data samples but still provides useful information to the model trainer, such as using techniques like noise addition, feature (instead of data) sharing, synthetic data generation, and other privacy-preserving methods on top of `Mycroft`. For a detailed discussion of extensions towards privacy-preserving data sharing, see Appendix G. Additionally, exploring versions of `Mycroft` that are resistant to strategic, and even malicious, data owners is a promising area for future work.

We hope our approach in this paper lays the foundation for more efficient and privacy-focused data-sharing frameworks, ultimately democratizing the training of highly performant ML models.

**Reproducibility Statement**

We have taken several steps in order to ensure reproducibility of our work. Firstly, we open source the code for all our experiments anonymously at https://anonymous.4open.science/r/Mycroft-73FE/ . This code includes the methodologies used for processing the datasets and running all our experiments. We use a mix of public and newly curated datasets which we have detailed in Section 4. We will also be releasing the Dogs & Wolves dataset we have curated to the community. Moreover, we have included details about the compute requirements of our method in Appendix D and and Section 4.3.

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

## A  SYMBOLS AND NOTATIONS

| Symbol | Description |
|---|---|
| DO | Data Owner |
| MT | Model Trainer |
| $D^{\text{MT}}$ | MT's dataset |
| $M_{\text{MT}}$ | MT's model |
| $D^{\text{test}}$ | MT's test dataset |
| $D_i$ | $i^{th}$ DO's dataset |
| $D_i^{\text{useful}}$ | Subset of $D_i$ retrieved by `Mycroft` or `random-sampling` |
| $D^{\text{hard}}$ | Subset of $D^{\text{test}}$ which is incorrectly classified and is shared with the DOs |

Table 5: Table of notations used in the paper.

## B  PROOFS

In this section, we prove the weak submodularity of the function obtained by subtracting the error function in Eq. 3 from the maximum value of the loss. That is, we need to prove that the following function is weakly submodular with paramter $\gamma'$:

$$f_{\lambda_1,\lambda_2}(\mathbf{w}) = L_{\max} - \min_{\|\mathbf{w}\|_0 \leq k} \left\| \sum_{z_j \in D_i} \mathbf{w}_j \nabla_{\theta_i} L(z_j) - \nabla_{\theta_i} L(D^{\text{hard}}) \right\| + \lambda_1 \|\mathbf{w}\|_2^2 + \lambda_2 \|\Psi\mathbf{w}\|_2^2, \quad (3)$$

under the conditions specified in Theorem 3.1. Given that this function is submodular, then the use of the Orthogonal Matching Pursuit (OMP) algorithm from Elenberg et al. (2016) will return a $k$-sparse subset with performance that is a $1 - e^{\lambda'}$ approximation of the maximum value.

*Proof of Theorem 3.1.* From Elenberg et al. (2016), a function is $\gamma'$ weakly submodular with $\gamma' \geq \frac{m}{M}$ where $m$ is the restricted strong concavity parameter and $M$ is the restricted smoothness parameter.

To prove that $f_{\lambda_1,\lambda_2}(\mathbf{w})$ is strongly concave with parameter $m$, we need to show that

$$-\frac{m}{2}\|\mathbf{v} - \mathbf{w}\|_2^2 \geq f_{\lambda_1,\lambda_2}(\mathbf{v}) - f_{\lambda_1,\lambda_2}(\mathbf{w}) - \langle \nabla f_{\lambda_1,\lambda_2}(\mathbf{w}), \mathbf{v} - \mathbf{w} \rangle \quad (4)$$

Plugging in $f_{\lambda_1,\lambda_2}(\cdot)$ from Eq. 1, we get

$$-\frac{m}{2}\|\mathbf{v} - \mathbf{w}\|_2^2 \geq -\lambda_1\|\mathbf{v} - \mathbf{w}\|_2^2 - \lambda_2\|\Psi\mathbf{v} - \Psi\mathbf{w}\|_2^2$$

$$\geq -\lambda_1\|\mathbf{v} - \mathbf{w}\|_2^2 - \lambda_2\|\Psi\|_2^2\|\mathbf{v} - \mathbf{w}\|_2^2,$$

where the final inequality arises from the property of the induced norm with respect to a matrix and $\|\Psi\|$ is the spectral norm of the distance matrix $\Psi$. This implies $m \leq 2(\lambda_1 + \lambda_2 \|\Psi\|_2^2)$.

To prove that $f_{\lambda_1,\lambda_2}(\mathbf{w})$ is restricted smooth with parameter $M$, we need to show that

$$f_{\lambda_1,\lambda_2}(\mathbf{v}) - f_{\lambda_1,\lambda_2}(\mathbf{w}) - \langle \nabla f_{\lambda_1,\lambda_2}(\mathbf{w}), \mathbf{v} - \mathbf{w} \rangle \geq -\frac{M}{2}\|\mathbf{v} - \mathbf{w}\|_2^2 \tag{5}$$

Expanding the term on the L.H.S. again, we get,

$$- \lambda_1 \|\mathbf{v} - \mathbf{w}\|_2^2 - \lambda_2 \|\Psi\|_2^2 \|\mathbf{v} - \mathbf{w}\|_2^2 - \sum_j \mathbf{v}_j (\sum_k (\mathbf{w}_k - \mathbf{v}_j) \nabla_{\theta_i}(z_j)^\mathsf{T} \nabla_{\theta_i}(z_k))$$

$$\geq -\lambda_1 \|\mathbf{v} - \mathbf{w}\|_2^2 - \lambda_2 \|\Psi\|_2^2 \|\mathbf{v} - \mathbf{w}\|_2^2 - k\nabla_{\max}^2 \|\mathbf{v} - \mathbf{w}\|_2^2,$$

where the final inequality arises from the $k-$ sparse condition on the weight vectors and the bound on the gradients of the loss function. This gives $M \geq 2(\lambda_1 + \lambda_2 \|\Psi\|_2^2 + k\nabla_{\max}^2)$.

Together, this gives $\gamma' \geq \frac{\lambda_1 + \lambda_2 \|\Psi\|_2^2}{\lambda_1 + \lambda_2 \|\Psi\|_2^2 + k\nabla_{\max}^2}$.

$\square$

## C  PSEUDO CODE AND RUNTIME ANALYSIS FOR MYCROFT

### C.1  PSEUDO CODE FOR MYCROFT

---
**Algorithm 2** OMP
---
**Require:** $D^{\text{hard}}$, DO's loss function : $L$, $D_i$, $M_i$'s parameteres $\theta$, regularization coefficients: $\lambda_1, \lambda_2$ , subset size: $k$, tolerance: $\epsilon$
1: $\mathcal{X} \leftarrow \emptyset$
2: $r \leftarrow \nabla_{\theta_i} L(D^{\text{hard}})$
3: **while** $\mathcal{X} \leq k$ and $r \geq \epsilon$ **do**
4: $\quad m \leftarrow \arg\max_j |Proj(\nabla_{\theta_i} L(D_i), r)|$
5: $\quad \mathcal{X} \leftarrow \mathcal{X} \cup \{m\}$
6: $\quad w^* \leftarrow \arg\min_w e'_{\lambda_1,\lambda_2}(w, \mathcal{X})$
7: $\quad r \leftarrow r - Proj(\mathcal{X}, w^*)$
8: **end while**
9: **return** $\mathcal{X}$, w
---

---
**Algorithm 3** FeatureSimilarity
---
**Require:** $D^{\text{hard}}$, $D_i$, Budget $k$,
1: $\phi(D^{\text{hard}}), \phi(D_i^{\text{useful}}) \leftarrow$ DO runs **FeatureExtractor**$(D^{\text{hard}}, D_i)$ $\qquad \triangleright$ **Unicom** or **Binnning**
2: $\Psi \leftarrow$ **ComputeDistances**$(\phi(D^{\text{hard}}), \phi(D_i^{\text{useful}}))$
3: $D_i^{\text{useful}} \leftarrow$ **RetrieveTopK**$(\Psi, k)$
4: **return** $D_i^{\text{useful}}$
---

### C.2  COMPLEXITY & RUNTIME OF MYCROFT

**Image datasets:** Mycroft for the image domain consists of two techniques: Unicom and Grad-Match. Here, we discuss the computation and memory complexity of both these techniques in order to give a sense of their efficiency and practicality. Unicom operates by projecting all data points in the representation space of the Unicom model, which is based on CLIP Radford et al. (2021), and computing distances between those data points. Thus, its compute and memory requirement scale in proportion to the number of data points to be projected as each sample requires a forward pass through the model to acquire its feature representation which needs to be held in stored for computing distances with other data points. Emprically, we find that this procedure takes less than 5 minutes and requires less than 4 GB of GPU memory for each experiment we present in this paper. GradMatch

---

**Algorithm 4** BinningDistance

---

**Require:** $D^{\text{hard}}$, DO, percentage to sample from DO: $r$, minimum number of non-empty bins for each feature: $b$, binning candidates list (in increasing order): $candidates$, features used for binning: $features$

1: $\mathcal{X}^{\text{DO}}, \mathcal{X}^{D^{\text{hard}}} \leftarrow$ ExtractBinningFeatures($D^{\text{hard}}$, DO, $r, b, candidates, features$)
2: $D \leftarrow \emptyset$
3: **for** $p^{\text{DO}}$ in $\mathcal{X}^{\text{DO}}$ **do**
4:     $d\_l \leftarrow \emptyset$
5:     **for** $p^{D^{\text{hard}}}$ in $\mathcal{X}^{D^{\text{hard}}}$ **do**
6:         $d \leftarrow$ GetDistance($p^{\text{DO}}, p^{D^{\text{hard}}}$)
7:         $d\_l \leftarrow d\_l \cup \{d\}$
8:     **end for**
9:     $D \leftarrow D \cup \{d\_l\}$
10: **end for**
11: **return** $D$

---

---

**Algorithm 5** ExtractBinningFeatures

---

**Require:** $D^{\text{hard}}$, DO, percentage to sample from DO: $r$, minimum number of non-empty bins for each feature: $b$, binning candidates list (in increasing order): $candidates$, features used for binning: $features$

1: DO$^{\text{samples}} \leftarrow$ sample(DO, $r$)
2: UnionSamples $\leftarrow D^{\text{hard}} \cup$ DO$^{\text{samples}}$
3: $\mathcal{X}^{\text{DO}} \leftarrow \emptyset$
4: $\mathcal{X}^{D^{\text{hard}}} \leftarrow \emptyset$
5: **for** $f$ in $features$ **do**
6:     NumBin $\leftarrow \max(candidates)$
7:     **for** $n$ in $candidates$ **do**
8:         NumFilled $\leftarrow$ CountNonEmptyBins(UnionSamples[$f$], $n$)
9:         **if** NumFilled $\geq b$ **then**
10:             NumBin $\leftarrow b$
11:             break
12:         **end if**
13:     **end for**
14:     edge$^f \leftarrow$ GetEdge(UnionSamples[$f$], NumBin)
15:     $\mathcal{X}^{\text{DO}} \leftarrow \mathcal{X}^{\text{DO}} \cup$ GetBinningCoordinates(DO[$f$], edge$^f$)
16:     $\mathcal{X}^{D^{\text{hard}}} \leftarrow \mathcal{X}^{D^{\text{hard}}} \cup$ GetBinningCoordinates($D^{\text{hard}}$[$f$], edge$^f$)
17: **end for**
18: **return** $\mathcal{X}^{\text{DO}}, \mathcal{X}^{D^{\text{hard}}}$

---

requires computing gradients for each data point in $D^{\text{hard}}$ and `DO`'s using the `DO`'s model once. In practice, we only use the gradients of the last two layers, which significantly reduces the compute and memory requirements. The gradients are then used to run the OMP algorithm which has a complexity of $\mathcal{O}(NM + Mk + k^3)$ for each of the $k$ iterations where $k$ is $|D_i^{\text{useful}}|$, $M$ is the dimension of the gradients and $N$ is $|D_i|$. For experiments in this paper, each experiment ran in under 10 minutes and required approximately 6 GigaBytes of memory.

**Tabular dataset:** The runtime for tabular dataset as described in 4 is $\mathcal{O}(MNF)$ where $M$ is the number of samples in `DO`, $N$ is the number of samples in $D^{\text{hard}}$, $F$ is the number of features used for *ExtractBinningFeatures*. Empirically, each experiment takes less than 5 minutes and requires less than 3 GB of memory.

## D FURTHER SETUP DETAILS

`MT` will evaluate the utility of `DO`'s data based on the framework found in Figure 3. Further details about the datasets and training process used to obtain the results in the main body of the paper are provided in this section.

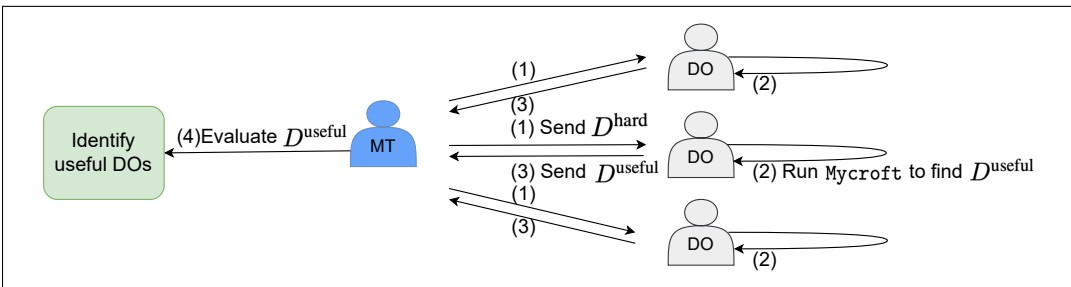

Figure 3: Framework for `MT` to evaluate the utility of `DO`'s data.

### D.1 DATASETS

#### D.1.1 DOGS & WOLVES DATASET

Neural networks are known to learn spurious correlations in supervised settings. While test data containing the correlations learnt during training often gets classified correctly, data which does not contain such correlations is prone to misclassification. We exploit this phenomenon to create a dataset which helps us simulate a controlled `MT`-`DO` interaction. In our case, the `MT` has a training and validation dataset which contains spurious correlations but a test dataset which does not contain them and thus their model suffers on the test dataset.

Concretely, we curate a dataset which consists of two classes: Dogs and Wolves. Spurious correlations are introduced in it by controlling the background of each image which can either be snow or grass. The `MT` has data from both animals being on one type of background. In particular, the dogs are on grass and the wolves are on snow. In the absense of negative examples, the model takes a shortcut by associating the true label with the background and not the animal. We refer to `MT`'s training and validation subset as Dogs & Wolves - Spurious. However, the model performs poorly when the test samples do not contain the spurious correlations i.e., dogs on snow and wolves on grass. We refer to this subset as Dogs & Wolves - Natural. We simulate a `DO` which has a dataset containing both Dogs & Wolves - Spurious and Dogs & Wolves - Natural and thus their model does not learn background related spurious correlations. An illustration of this dataset is provided in Figure 5.

Now, if the `MT` wants to perform well on data from Dogs & Wolves - Natural, they must acquire data from that distribution and retrain their model in order to break the spurious correlations. This motivates the `MT` to acquire new data form the `DO`. It should be noted that the `MT` is oblivious as to why their model performs poorly on Dogs & Wolves - Natural.

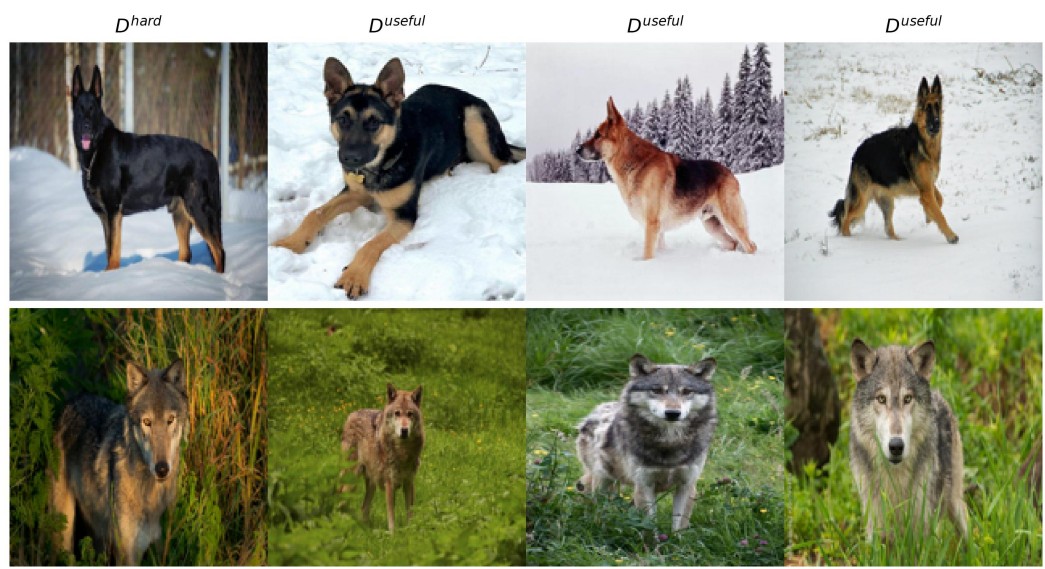

Figure 4: Top-k retrieved $D_i^{\text{useful}}$ samples using Unicom for $D^{\text{hard}}$ from the Dogs & Wolves dataset.

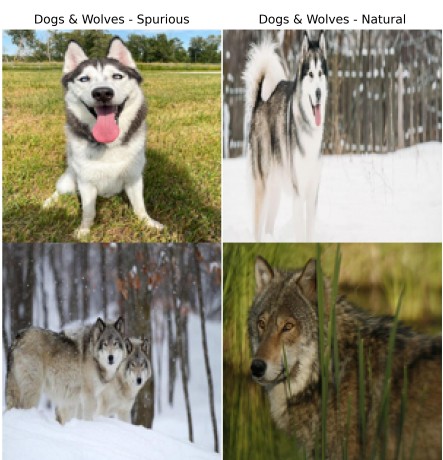

Figure 5: Subsets in the Dogs & Wolves dataset. The first column shows Dogs & Wolves - Spurious where the dogs are on a grass background and the wolves are on snow. The second column shows Dogs & Wolves - Natural where the dogs are on snow and the wolves are on grass.

### D.1.2  FOOD DATASET

Food-101 contains 101,000 images of 101 food classes with 750 and 250 images for each category for training and testing. UPMC Food-101 is a twin dataset to Food-101 and thus has the same number of categories and size as Food-101. ISIA Food-500 is another food recognition dataset with approximately 400,000 images for 500 food classes and has many classes which intersect with the set of classes in UPMC Food-101. In our experiments, we use Food-101 as the MT's dataset and UPMC Food-101 and ISIA Food-500 as datasets of two different DO's.

### D.1.3  TABULAR DATASET

**Data source:**
The dataset we use consists of five captures (scenarios) of different IoT network traffic Garcia et al. (2020). Each network consists of traffic of two types : benign traffic (when the IoT devices are not not under attack) and malicious traffic (when the devices are under attack). The attacks are executed

in a Raspberry Pi and each capture can suffer from different attacks. Details about the types of benign/attack present in each capture can be found in Table 6.

Table 6: Attacks present in each capture, Benign stands for Benign traffic. PHP stands for Part Of A Horizontal PortScan attack, CC stands for C&C attack, CCT stands for C&C Torii.

| Capture | Benign/Attacks present |
|---------|------------------------|
| 1       | Benign, PHP,           |
| 3       | Benign, CC, PHP        |
| 20      | Benign, CCT            |
| 21      | Benign, CCT            |
| 34      | Benign, CC, PHP        |

**Data Labeling:**
From the raw pcap files (which contain the raw network traffic data), we use the Python library NFStream Listed (2024) to extract feature flows (in tabular format) from pcap files. We then match the timing of the flow with the timing that the attack was executed as mentioned in the data source Garcia et al. (2020) to label the data. After labelling the flows, we split these flows based on whether they are benign or malicious based on their attack type.

**Model Trainers:** We define an MT to be the IoT device in the captures that want to improve its model's ability to predict some particular attack. In this study, we have 7 different MTs. In addition, because the attack data in this dataset has quite uniform distribution (most likely because they are conducted in a lab-setting) such that if the model has been trained on the attack, they are very likely to predict a future attack of the same type with high accuracy, we assume that these MTs are only trained on benign data. In reality, this scenario is possible because if the network is new, the chances of them being trained on attack data for this new network is low. The MTs see a very small number of attack data which its model fail to predict and would like to get more data from DOs to improve their models.

**Data Owners:** We artificially inflate the number and complexity of DOs by mixing data from different captures to generate 95 new DOs. This will increase the difficulty of finding relevant samples in DOs and simulate the real scenarios where DO are often quite complex.

**Details for $D^{\text{hard}}$:** The quantity of $D^{\text{hard}}$ which each MT possesses is very small (2% of the malicious data) and is chosen randomly from the malicious data of the MT's dataset.

### D.2 TRAINING DETAILS

Here, we provide more details about the training procedure we used for obtaining the neural networks we use for our computer vision tasks.

For the public image datasets we use, we observe that the model performs well on most classes perform and thus, there is little to gain from external data augmentation. Therefore, to simulate a more realistic setting, we reduce performance on certain classes of the MT's model by training them with limited training data. On average, we use 10% of the original training data for the classes we choose to augment using external datasets and attain an average accuracy of 65% for them.

The MT's and DO's models are ResNet50 models pretrained on Imagenet and finetuned on their respective datasets. The MT's and DO's base model is trained for 120 epochs using a learning rate of 0.03 with a cosine annealing weight decay. The MT's augmented models, $M'_{\text{MT}}$, are obtained by finetuning their base models for 25 epochs on $D_i^{\text{useful}}$ and their original training dataset.

## E ADDITIONAL RESULTS & ABLATION STUDIES

Here, we evaluate which training phase of the DO's model provides the most useful gradient information for data selection. For evaluation, we chose the Dogs & Wolves dataset as the DO and perform gradient matching for several checkpoints. We compare the percentage of samples in the retrieved

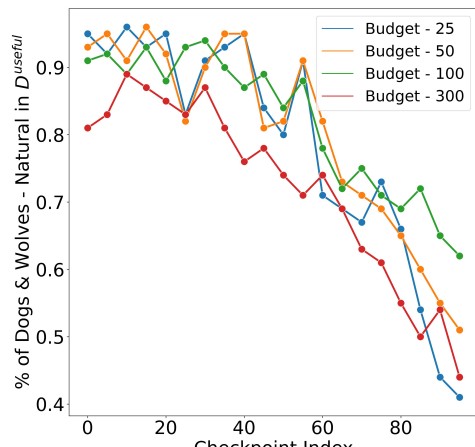

Figure 6: Effect of the checkpoint used for GradMatch on $D_i^{\text{useful}}$. Earlier checkpoints tend to provide more useful gradient information for the OMP algorithm.

subset which belong to the Dogs & Wolves - Natural data subset since those are the only useful samples in the `DO`'s dataset. We plot the results in in Figure 6 and observe that earlier checkpoints indeed provide more useful gradients.

### E.1 IMAGE DATASETS

We conduct ablation study on image datasets to see how the selection of checkpoints affect Grad-Match's ability to select $D_i^{\text{useful}}$. We find that earlier checkpoints tend to provide more useful gradient information for the OMP algorithm. Refer to Figure 6 for the results.

### E.2 TABULAR DATA

For this dataset, as noted in §4.2, the results presented only use feature similarity. We discuss the reasons here. First, the models that perform the best on this dataset are tree-based classifiers for which gradient matching does not apply. In addition, to resemble the effect of gradient matching for tabular data, we have also attempted to retrieve $D_i^{\text{useful}}$ based on `DO`'s model confidence score and DecisionTree's decision path when trained on `DO`'s data. We find that the performance of these approaches are not as good as `Mycroft`. Potential reasons for why these approaches do not work are (i) features that differentiate benign and malicious traffic for `DO` might not be features that are important for `MT`'s model (as can be seen in the fact that for some `DO`s, `MT`'s model does not improve after data sharing) and (ii) the DecisionTree's decision path when trained on `DO`'s data might be over-reliant on only one or very few features, hence, do not provide useful signals to select $D_i^{\text{useful}}$.

**Performance with different classifiers:** In this section, we present `MT`'s F1 score after data sharing using `random-sampling` and `Mycroft` for different classifiers. We find that DecisionTree seems to be the best classifier for this dataset. Refer to Figure 7, 8 and 9 for the results.

**Performance when $D_i^{\text{useful}}$ is selected based on different data selection algorithm:** To explore whether DecisionTree's decision path can be used to select $D_i^{\text{useful}}$, we use $D_i^{\text{useful}}$ budget of 5 samples and compare `MT`'s F1 score after data sharing when $D_i^{\text{useful}}$ is retrieved from samples of the same decision path as $D^{\text{hard}}$, of different decision paths as $D^{\text{hard}}$, retrieved from `random-sampling` and `Mycroft`. Note that to make this study comparable, we only consider cases where samples of the same decision path as $D^{\text{hard}}$ and samples of different decision path as $D^{\text{hard}}$ can be found. This total up to 466 cases. We find that although sharing samples of same decision paths as $D^{\text{hard}}$ can be slightly better than `random-sampling`, `Mycroft` still outperforms this approach significantly.

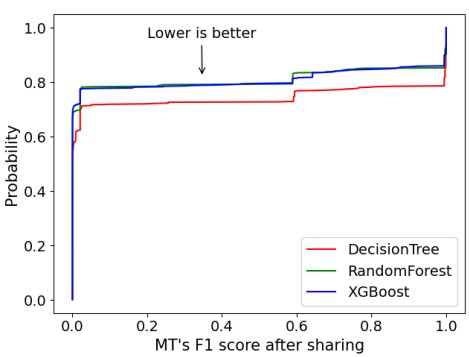 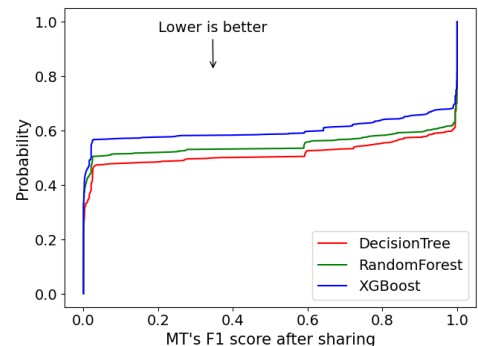

(a) CDF of MT's F1 score after data sharing using `random-sampling` with $D_i^{\text{useful}}$ budget of 5 samples for different classifiers. DecisionTree seems to be the best classifier for this dataset.

(b) CDF of MT's F1 score after data sharing using `random-sampling` with $D_i^{\text{useful}}$ budget of 100 samples for different classifiers. DecisionTree seems to be the best classifier for this dataset.

Figure 7: Performance of `random-sampling` for different classifiers for the tabular dataset.

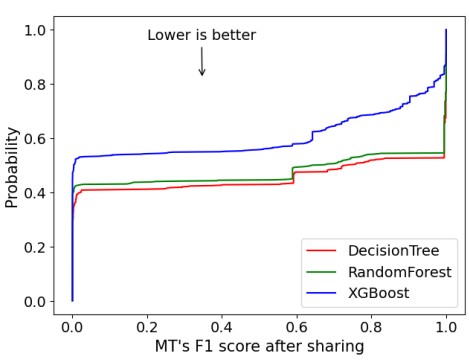 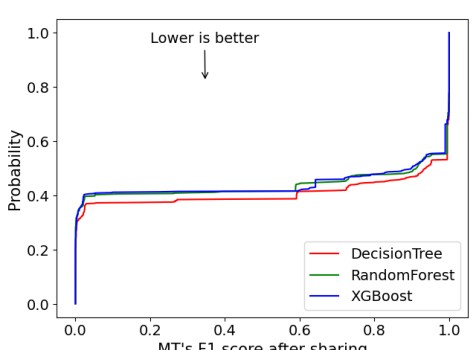

(a) CDF of MT's F1 score after data sharing using `Mycroft` with $D_i^{\text{useful}}$ budget of 5 samples for different classifiers. DecisionTree seems to be the best classifier for this dataset.

(b) CDF of MT's F1 score after data sharing using `Mycroft` with $D_i^{\text{useful}}$ budget of 100 samples for different classifiers. DecisionTree seems to be the best classifier for this dataset.

Figure 8: Performance of `Mycroft` for different classifiers for the tabular dataset.

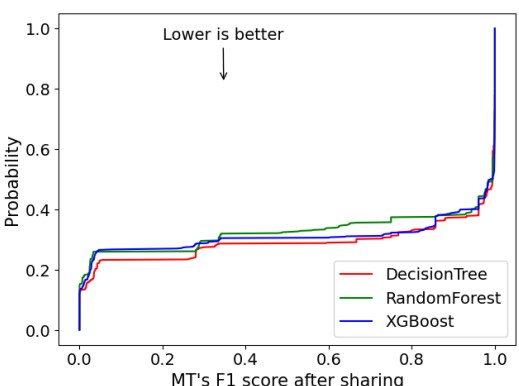

Figure 9: CDF of MT's F1 score after data sharing for `full-information` for different classifiers. DecisionTree seems to be the best classifier for this dataset.

Refer to Figure 10 for the results.

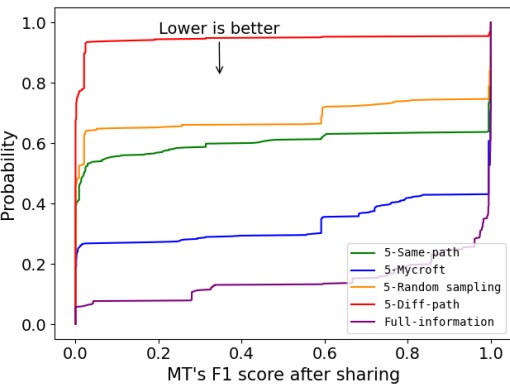

Figure 10: CDF of MT's F1 score after data sharing using different data selection methods with $D_i^{\text{useful}}$ budget of 5 samples and using `full-information`. Classifier is DecisionTree. N = 466 cases where samples of the same decision path as $D^{\text{hard}}$ and samples of different decision path as $D^{\text{hard}}$ can be found.

**Performance of combining Mycroft and other data selection methods:** To explore the effects of combining Mycroft and other data selection methods such as DecisionTree's decision path and `random-sampling`, we let $D_i^{\text{useful}}$ to be made up of samples selected by Mycroft and samples selected by these other data selection methods. We find that MT's F1 score after data sharing is not significantly improved when combining Mycroft with other data selection methods compared to using Mycroft alone. Refer to Figure 11 for an example of the results.

**Performance using different distance metrics:** Depending on the type of data (image versus tabular) and the dataset, different distance metrics might be chosen to give better performance. For example, for the tabular dataset, after experimenting with different distance metrics such as total variance distance Verdú (2014), autoencoder distance Akrami et al. (2020), etc, we found that existing distance metrics do not give good performance. As such, we come up with a simple, intuitive distance metrics called "binning distance" to measure the distance between $D^{\text{hard}}$ and DO's samples. The intuition behind this distance metrics is that because different features of tabular data has different units and scales, we first bin the data (using histogram of uniform distances) based on each feature's own distribution and then calculate the distance between $D^{\text{hard}}$ and DO's samples based on the bins. Because the distribution of the same feature might be different for DO's data and $D^{\text{hard}}$, we need to based our binning method on the feature distribution of both sources of data to ensure that the

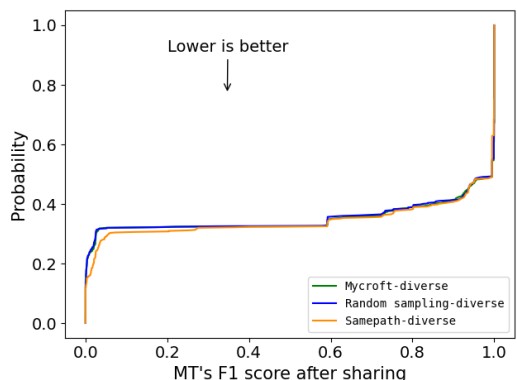

Figure 11: CDF of `MT`'s F1 score after data sharing when $D_i^{\text{useful}}$ is made up of 100 samples retrieved from `Mycroft` and 5 samples retrieved by different data selection methods. Classifier is DecisionTree. N = 574 cases where samples of the same decision path as $D^{\text{hard}}$ can be found. Note that the 5 samples later selected by each data selection method must be different from 100 samples already selected by `Mycroft`.

binning is not over-fitted to one source of data. As such, we first create $D^{\text{base}}$ which is the union of $D^{\text{hard}}$ and some of `DO`'s samples and then bin the data based on the feature distribution of $D^{\text{base}}$. We then calculate the distance between $D^{\text{hard}}$ and `DO`'s samples based on the bins. We find that this distance metrics gives better performance compared to other distance metrics. Refer to Table 4 for the algorithm to calculate the binning distance.

# F  CASE STUDIES

Here, we present the details for the `DOs` and `MT`'s used for the Scenario 3 in Section 5.

**Image datasets:**

`DO-1`: This `DO` contains the highest quantity of data from the same distribution as $D^{\text{hard}}$ and is the same as the `DO` we use in other experiments involving the Dogs & Wolves dataset. This `DO` should provide the highest utility to the `MT`.

`DO-2`: `DO-2` is a noisy version of `DO-1` where we introduce noise by randomly transforming the images using PyTorch transforms tor. The transforms we apply include Random crops, resizing, flipping, changing contrast and perspective. We expect the utility of training on such images to be lower as compared to the clean images.

`DO-3`: `DO-3` has randomly sampled data from dog and wolf classes in the ImageNet dataset. We empirically verify that it contains a subset of data from the distribution required by the `MT` and will thus be useful to the `MT` to some degree.

`DO-4`: This `DO` contains a small subset of the useful samples contained in `DO-1`. While useful in nature, this `DO`'s ability to signal its utility should be limited.

`DO-5`: `DO-5` contains no data from the required training distribution and only consists of the data from `MT`'s training distribution. This type of data should have the least utility.

**Tabular datasets:**

In this case study, we focus on the `MT`-7, which belongs to capture 20 and trying to predict the C&C Torii attack. The goal of the `MT` is to select, amongst 95 potential `DOs`, the ones that will contain useful data that helps predicting this attack. If performing `full-information` with `DOs`, 88 out of 95 `DOs` will give `MT` an F1 score of > 0.99% while the remaining 7 `DOs` will give `MT` an F1 score of <= 0.0%. Given that the utility of useful `DOs` is almost the same, the goal of `MT`-7 is is mostly to retrieve useful `DOs` rather than rank them because any useful `DOs` can improve `MT`'s performance.

We find that with a $D_i^{\text{useful}}$ budget of only 5 samples, `Mycroft` can retrieve 92 useful `DOs` that can give `MT` an F1 score of > 0.99% (the rest gives `MT` an F1 score of <= 0.0%). This is much better compared to `random-sampling` which only retrieves 25 useful `DO` that give `MT` an F1 score of >

0.99% (the rest give `MT` an F1 score of $< 0.0\%$). Examining the cases where `Mycroft` outperforms `full-information` in selecting useful `DO`, we find that `Mycroft` can retrieve useful `DO`s while `full-information` is unable to because irrelevant data affects `full-information`'s ability to retrieve useful samples. For example, there is a `DO` created by mixing samples of PartOfAHorizontalPortScan attack from capture 3 with samples of C&C attack from capture 3. For this `DO`, `full-information` will gives `MT` an F1 score of 0.0% while `Mycroft` gives `MT` an F1 score of 0.99%. This is because although C&C attack samples from capture 3 are useful for `MT` (give `MT` an F1 score of $> 0.99\%$) and PartOfAHorizontalPortScan attack samples from capture 3 are not (give `MT` an F1 score of $< 0.0\%$). `full-information` will give `MT` an F1 score of 0.0% because the irrelevant PartOfAHorizontalPortScan attack samples from capture 3 will dominate useful C&C attack samples data from capture 3. `Mycroft` is able to identify this `DO` as useful because it can select only the useful C&C attack samples from capture 3 and ignore irrelevant PartOfAHorizontalPortScan attack samples from capture 3.

# G  ADDRESSING PRIVACY CONCERNS OF DATA SHARING

This section demonstrates the feasibility of external data augmentation for ML problems of practical interest. Since data sharing raises several concerns with regards to the privacy of the data, we discuss some strategies to mitigate privacy concerns as well as some alternative techniques to `Mycroft` and why they are not feasible for solving our problem.

**Why not use federation?:** The main issue with methods such as federated learning, whether privacy-preserving or not, is the need to federate. There needs to be a centralized server which collects the model updates from different agents, aggregates them, and then sends them back for the next round. In many scenarios where `MT` needs to improve its model performance, it may not have the resources or technical ability to set-up this centralized server. It would also need to convince the `DO`s to participate in this federated learning protocol for a number of rounds. In contrast, external data augmentation does not need a centralized coordinating server, just a method to send and receive data asynchronously. Completely decentralized peer-to-peer learning is also an option, however this needs the model that `MT` wants to train to be passed around in some fashion to its peers, which violates constraint C1.

**What about secure multi-party computation?:** The main issue with using MPC to solve the problem tackled in this paper is that if `DO` computes the function (model) jointly with `MT`, it will likely use all of its data when training. Such a protocol will be expensive and can leak information Salem et al. (2018) about all of `DO`'s' training data. This motivates this paper's focus on `DO` only sharing a part of its data.

Given these barriers to the adoption of direct decentralized model training, we believe that a good solution to the problem posed in this paper is using external data augmentation. Mycroft provides an initial protocol that has been demonstrated to be useful while sharing data within a specified budget. However, even this may be too much information to share in certain settings. There are several ways in which the privacy of Mycroft can be improved in follow-up work, which we discuss here:

1. Sharing features instead of data: `MT` can, instead of directly sharing data with `DO`, just share an appropriate compressive feature representation of the data which reduces the amount of information being shared. This will make it harder to determine the exact data held by `MT`. However, this can leak information about `MT`'s model, as well as cause issues with data selection if there is mismatch between the architecture of the models used by `MT` and `DO`, since comparisons will have to be done in feature space directly to determine utility.

2. Adding noise to the data: Differentially private noise addition can be used to prevent the inference of sensitive attributes from $D^{\text{hard}}$, which is particularly important for domains such as electronic health records. As is normal, this will lead to a privacy-utility trade-off for the identification of $D^{\text{useful}}$, which is an interesting direction for future work. Noise-resilient protocol design for data selection may be challenging depending on the magnitude of noise added.

3. Operating on encrypted data: The most computationally expensive method for ensuring privacy would be for one or both of `MT` and `DO` to only share encrypted data. Past work Bost et al. (2014) has shown that classification can be performed over encrypted data using primitives such as partially homomorphic encryption. The proposed data selection and subsequent fine-tuning could

be extended to operations only over encrypted data, although both efficiency and effectiveness are likely to suffer.

4. Using ZKPs: In case the DO does not wish to share data at all, the following thought experiment implies that ZKPs may be a fruitful approach for a DO to prove to MT that it does have relevant data, once it has received $D^{\text{hard}}$ and used an appropriate data selection mechanism. A possible protocol would work as follows: 1) MT would share some information $I$ about $D^{\text{hard}}$ (such as an encrypted sample of the data or a sample with noise added); ii) DO would use $I$ to find $D^{\text{useful}}$ from within its own dataset, using Step 3 in Algorithm 1 (DataSelect); iii) DO would then commence a ZKP protocol with MT to prove the statement that 'DO has data that is useful for MT, without ever sharing its data directly. While this prevents MT from training a model on an augmented dataset, it does allow for the determination of which DOs are useful and which not. Note that MT does have to share some information about $D^{\text{hard}}$ in order to have a useful protocol. Designing such a zero-knowledge proof protocol is beyond the scope of this paper, although the fundamental abstraction that motivates Mycroft would remain the same for the overall algorithm, as would the use of some DataSelect. Hardware solutions such as trusted enclaves at the DOs' end could be used to secure $D^{\text{hard}}$ as well.

