# OpenReview forum: "Mycroft: Towards Effective and Efficient External Data Augmentation"
_ICLR.cc/2025/Conference — Submitted to ICLR 2025_

### Official Review · Reviewer_ra4n · 2024-11-03

**Soundness:** 2
**Presentation:** 2
**Contribution:** 1
**Rating:** 3
**Confidence:** 3

**Summary:**

This study proposes the problem of external data augmentation where model trainers aim to acquire data from outside sources without sharing their own data. After defining this problem, the study proposes two algorithms and their combination as potential solutions for data owners to provide effective data for model trainers. The algorithms are based on gradient matching and feature similarity, two commonly used techniques throughout the prior literature on this topic. Finally, the study aims to demonstrate the effectiveness of these solutions to addressing the formalized problem on tabular and image datasets.

**Strengths:**

- The paper takes an often discussed problem and places it on formal ground and demonstrates how existing solutions can be used to address this problem.
- The empirical results demonstrate the overall utility of being able to source additional data from external data sources
- The FuncFeat algorithm provides a simple and what appears to be effective way of combining two commonly used data valuation techniques

**Weaknesses:**

- There is not much in this paper that has not already been presented in other papers. I see this paper as more of a formalizaton of a problem and then a survey of the existing techniques for that problem. While that is useful it is not substantial enough for conference publication.
- It would be helpful for the authors to lay out exact real life applications where this setup is happening / being suggested
- Overall, I think the results are interesting and in line with what prior works have demonstrated in terms of using these algorithms for the outlined problem.

**Questions:**

1. What makes the formalized problem and approaches provided different from the data valuation literature?

---

### Official Review · Reviewer_1L6m · 2024-11-04

**Soundness:** 3
**Presentation:** 3
**Contribution:** 2
**Rating:** 3
**Confidence:** 3

**Summary:**

To tackle the data deficiency while completing new tasks, this paper proposes to evaluate the data utility from different sources called Mycroft. Specifically, the model trainer provide data samples for owners to select relevant data for a trainer. Then, the trainer evaluates and decides whether further requests are needed. Further experiments demonstrate the efficacy and robustness under the noise data scenario.

**Strengths:**

- This paper is easy to follow .

**Weaknesses:**

- It is not clear about the motivation of this paper, privacy protection, or others, it also lacks some real scenarios, it makes me feel like dataset condensation or distillation.
- One of the challenges in 2.1 is the reluctance to share data due to privacy, so why can the MySoft framework confirm that they are willing to share a small amount of data? There should still be privacy in there. These challenges are also related to FL.
- The problem this paper investigated do not seem to be very real needs
- What’s the performance of MySoft and some other dataset distillation methods, that DOs can distill their dataset into a subset that best represents the local data?
- This hypothesis assumes that MT has learned the information it needs to improve before, but has no good performance yet. If MT needs to test categories it has never seen before, is MySoft still useful. According to my understanding, when MT and DO data are similar, reasonable proxy can help select useful data. What will happen when new task data is needed or the domain gap between reasonable proxy and the original model is large?

**Questions:**

See weakness.

---

### Official Review · Reviewer_Tzx5 · 2024-11-04

**Soundness:** 2
**Presentation:** 2
**Contribution:** 2
**Rating:** 3
**Confidence:** 3

**Summary:**

The paper studies how model trainers could acquire a useful subset of data from data owners in a setting where both model trainers and data owners can share only a small amount of their data with each other due to regulatory and computational limitations. In particular, the authors study a protocol where a model trainer shares hard examples with data owners. Then, data owners select the most appropriate fixed-size subset of their data to share back with the model trainer.

The authors analyze this problem in the following way. Section 2 motivates the problem and formally introduces the data-sharing problem. To solve the data-sharing problem, the authors introduce Mycroft, a data-efficient subset selection method, in Section 3. For subset selection, Mycroft can utilize gradient-based (Section 3.1) or feature-based (Section 3.2) similarity metrics or FuncFeat (Section 3.3) loss. Section 4 empirically tests the method for image and tabular datasets. These experiments show that Mycroft is more efficient than random-sampling and that Mycroft can almost match the full-information baseline with a small data budget (Table 1 and Figure 1). Finally, Section 5 demonstrates that Mycroft could work in more general contexts where features might be corrupted, labels might be corrupted or missing, or the model trainer needs to choose between several providers.

**Strengths:**

- The problem of collaboration between model trainers and data owners is relevant in many contexts.
- The problem formulation, Equation (1), seems novel.
- The proposed approach, Mycroft, seems to solve the subset selection problem well (Section 4).
- Additionally, Mycroft seems to have additional regularization properties (Section 5).
- An observation that an earlier checkpoint is better for gradient matching (Section 4.3) is interesting.

**Weaknesses:**

- RetrieveTopK function in Algorithm 3 is confusing. The textual explanation suggests that this function is a sorting "greedy heuristic." However, it is hard to understand how this function is "ensuring coverage of $D^{hard}$."
- Algorithms 4 and 5 are hard to understand without textual description.
- The inequality below Equation (5) in the proof of Theorem 3.1 does not match the definition of $e_\lambda$ (Equation (1)). From what I see, the proof suggests that the first gradient-matching term in Equation (1) is squared.
- I do not clearly understand how sparsity implies the inequality below Equation (5).
- The definition of $f_{\lambda}$ in the proof of Theorem 3.1 does not match the definition of $f_{\lambda}$ in Section 3.
- The description of OMP by Elenberg et al. (2016, Algorithm 3) does not result in Algorithm 2 because Algorithm 2 does not account for the gradient of the feature-regularization term, $\lVert \Psi w \rVert^2$.
- Section 4 does not specify the size of $D^{hard}$, which makes the interpretation of the results hard.
- Sections 4 and 5 evaluate the final model only on the subset of $D^{hard}$. While I see how this metric reflects data relevance, it is also important to know how the additional data affects the performance on the whole dataset $D^{val}$ or $D^{test}$, especially for experiments in Section 5.
- Similarly, it is unclear in which experiments the final evaluation happens on the held-out portion of $D^{hard}$ and in which experiments the evaluation happens on the whole $D^{hard}$. Thus, it is impossible to know whether the method overfits to $D^{hard}$.
- The paper does not conduct experiments in settings where data privacy is protected (e.g., with any technique from Appendix G). Since privacy is a key concern in data sharing, judging the method's efficacy without such experiments is hard.

**Questions:**

- How does the RetrieveTopK function in Algorithm 3 work?
- Can the authors explain how they derive the inequality below Equation (5) in the proof of Theorem 3.1 in more detail?
- Why does Algorithm 2 in the paper differ from Algorithm 3 of Elenberg et al. (2016)?
- Is there some rationale behind the selection of the datasets for Section 4? Is there some relevant work that uses these datasets?
- What is the size of $D^{hard}$ in experiments?
- In which experiments does the evaluation occur on the held-out portion of $D^{hard}$?
- How might privacy protection quantitatively affect Mycroft's efficacy? Can the authors conduct some experiments with privacy protection techniques?

---

### Official Review · Reviewer_BcqH · 2024-11-10

**Soundness:** 4
**Presentation:** 3
**Contribution:** 3
**Rating:** 5
**Confidence:** 3

**Summary:**

The paper introduces "Mycroft," a method designed to enable model trainers to assess the utility of external data sources with limited data sharing, addressing privacy and budget constraints. The approach identifies relevant data subsets from data owners by evaluating both feature and functional similarity. Experimental results demonstrate that Mycroft approaches the performance of a full-information baseline with significantly reduced data sharing.

**Strengths:**

1.Clear Motivation: The paper addresses a real and relevant problem in machine learning, especially in settings where data sharing is constrained by privacy or budget.
2.Methodological Rigor: Mycroft leverages both gradient-based functional similarity and feature similarity to enhance data selection. The approach is well-supported by mathematical modeling and empirical evidence.
3.Comprehensive Experiments: The authors conduct thorough experiments across multiple domains (e.g., vision, tabular data) and explore various scenarios, including noise robustness and missing label settings.
4.Practical Utility: Mycroft offers a data-efficient solution, making it potentially valuable for smaller entities or applications where data access is restricted.

**Weaknesses:**

1.Limited Novelty in Data Selection Techniques: The core techniques (functional similarity via gradient matching and feature similarity) are adaptations of existing methods, which limits the novelty. While combining them is beneficial, this integration may not be sufficient to stand out as a fundamentally new approach.
2.Scalability Concerns: Although the authors provide complexity analyses, the scalability of Mycroft, especially with high-dimensional datasets, remains a concern. For larger datasets, the computational requirements for gradient-based data selection could become prohibitive.
3.Potential Vulnerability to Strategic Data Owners: The paper briefly mentions the possibility of strategic or malicious data owners, but it does not explore how Mycroft would handle cases where data owners intentionally provide misleading subsets.

**Questions:**

1.How does Mycroft perform in cases where Dhard is not well-defined or when it is difficult for the model trainer to identify misclassified examples?
2.Could data owners exploit Mycroft by strategically selecting non-informative samples within the constraints, and how might the protocol mitigate such behavior?
3.How would Mycroft perform on high-dimensional datasets where gradient-based data selection could be computationally expensive?

---

### Meta-Review · Area_Chair_Yc8V · 2024-12-19

**Metareview:**

The reviewers were not excited about the paper. Additionally the authors did not submit rebuttals to clarify the concerns of the reviewers. There are a bunch of problem formulation and algorithm description concerns that require addressing. We would recommend the authors to address those in future incarnations of the paper.

**Additional Comments On Reviewer Discussion:**

See above.

---

### Decision · Program_Chairs · 2025-01-22

Reject